# Study on the Synthesis of Nano Zinc Oxide Particles under Supercritical Hydrothermal Conditions

**DOI:** 10.3390/nano14100844

**Published:** 2024-05-12

**Authors:** Panpan Sun, Zhaobin Lv, Chuanjiang Sun

**Affiliations:** College of Mechanical & Electrical Engineering, Shaanxi University of Science & Technology, Xi’an 710021, China; lvzb@sust.edu.cn (Z.L.); 230511036@sust.edu.cn (C.S.)

**Keywords:** nanoparticles, crystallite size, zinc oxide particles, supercritical hydrothermal synthesis

## Abstract

The supercritical hydrothermal synthesis of nanomaterials has gained significant attention due to its straightforward operation and the excellent performance of the resulting products. In this study, the supercritical hydrothermal method was used with Zn(CH_3_COO)_2_·2H_2_O as the precursor and deionized water and ethanol as the solvent. Nano-ZnO was synthesized under different reaction temperatures (300~500 °C), reaction times (5~15 min), reaction pressures (22~30 MPa), precursor concentrations (0.1~0.5 mol/L), and ratios of precursor to organic solvent (C_2_H_5_OH) (2:1~1:4). The effects of synthesis conditions on the morphology and size of ZnO were studied. It was found that properly increasing hydrothermal temperature and pressure and extending the hydrothermal time are conducive to the more regular morphology and smaller size of ZnO particles, which is mainly achieved through the change of reaction conditions affecting the hydrothermal reaction rate. Moreover, the addition of ethanol makes the morphology of nano-zno more regular and significantly inhibits the agglomeration phenomenon. In addition to the change in physical properties of the solvent, this may also be related to the chemical bond established between ethanol and ZnO. The results show that the optimum synthesis conditions of ZnO are 450 °C, 26 MPa, 0.3 mol/L, 10 min, and the molar ratio of precursor to ethanol is 1:3.

## 1. Introduction

Zinc oxide nanoparticles are high-performance semiconductor inorganic compounds with many excellent properties, such as excellent optical, electrical performance, and catalytic performance [1]. Their low-cost, environmentally friendly, excellent-stability, and antibacterial properties make them widely used in fields such as solar cells [2], optoelectronic materials [2], light-emitting diodes [3], and photocatalysis [4]. Among many photocatalysts, ZnO has become the best alternative to TiO_2_. Compared with TiO_2_, the electron mobility of ZnO nanoparticles is 10–100 times higher [5]. At the same time, ZnO nanoparticles have higher quantum efficiency, better photocatalytic activity, and greater stability. Therefore, ZnO nanoparticles have developed into a highly promising optical material in the field of optoelectronic semiconductors and have been widely studied and applied in the industrial and scientific fields [6]. We can conclude that developing a simple and continuous manufacturing method for ZnO nanoparticles is of great significance.

Currently, various ZnO nanoparticle synthesis methods have been studied by researchers to obtain smaller ZnO nanoparticles. Smaller particles signify larger surface area, which gives them better catalytic activity and higher quantum efficiency. The traditional synthesis methods for zinc oxide nanoparticles are the direct precipitation method [7], sol-gel method [8], homogeneous precipitation method [9], etc. However, due to their high cost, long reaction time, and large product particles, they are not suitable for commercial production of ZnO. The supercritical hydrothermal synthesis method is a novel method used for preparing various kinds of nanomaterials [10,11,12]. To reach supercritical hydrothermal conditions, zinc salt precursor compounds are rapidly heated by directly contacting supercritical water, and zinc oxide nanocrystals are formed during the fast heating period [13]. Compared to traditional chemical synthesis methods, supercritical hydrothermal synthesis technology has several advantages. Firstly, it allows a simple reaction operation with a step reaction, making it quite suitable for industrial applications. Secondly, it is highly efficient and fast, with high reaction rates and crystallinity, and during extremely short reaction times, small and well-distributed products are obtained. Thirdly, it offers strong control over the synthesis, allowing for sufficient production of different particle sizes and crystal types by simple variation in synthetic conditions and appropriate solution stoichiometry. Lastly, it has low production costs compared to traditional methods [14]. At present, researchers have successfully synthesized zinc oxide nanorods [15], porous zinc oxide materials [16], and zinc oxide nanospheres and nanoflowers [17] through supercritical hydrothermal synthesis.

Mao Zhiqiang [18] et al. synthesized nano-zinc oxide with controllable morphology using the supercritical hydrothermal synthesis method and studied its photocatalytic performance. They elaborated on the process of preparing zinc oxide using the supercritical hydrothermal method, the crystal growth mechanism, and the influence of process parameters on zinc oxide particles. They concluded that the formation of zinc oxide crystals is mainly affected by the solubility of zinc oxide in the reaction conditions. With lower solubility, the production rate of zinc oxide usually increases. This is because low solubility means that more zinc oxide can be precipitated in supercritical water, providing more reactants to participate in the formation process of zinc oxide. Secondly, the formation of zinc oxide is also affected by Ostwald ripening and anisotropic growth during the growth process. Satoshi Ohara et al. [19] achieved continuous production of zinc oxide nanorods through the supercritical hydrothermal synthesis method. They obtained well-shaped, highly crystalline, and pure zinc oxide nanorods at 400 °C and 30 MPa. Their study found that higher-temperature conditions benefit the formation of zinc oxide nanorods. Under this condition, the decrease in solubility facilitates the formation of zinc oxide crystal cores, thereby accelerating the formation rate of zinc oxide nanorods.

Ludmila Motelica et al. [20] compared the properties of the ZnO nanoparticles obtained by solvolysis using a series of alcohols: primary from methanol to 1-hexanol, secondary (2-propanol and 2-butanol), and tertiary (tert-butanol). The results show that ZnO nanoparticles can be successfully synthesized in all primary alcohols, but the final product ZnO cannot be obtained by using secondary or tertiary alcohols, which emphasizes the importance of the solvent used. The shape of the obtained nano-zno particles depends on the alcohols used. The shape of nano-zno synthesized by different alcohols is different. ZnO synthesized in methanol is spherical, becomes polyhedral under 1-butanol, and becomes rod-like in 1-hexanol. In addition, Ludmila Motelica et al. [21] synthesized ZnO nanoparticles (NPs) using Zn(CH_3_COO)_2_·2H_2_O in alcohols with different numbers of −OH groups. The effects of different alcohol types (n-butanol, ethylene glycol, and glycerol) on the size, morphology, and properties of ZnO NPs were studied. The results show that one-step synthesis of ZnO nanoparticles is suitable for alcohols with only one or two hydroxyl parts. Moreover, the size of synthesized ZnO particles also depends on the type of alcohol used. Alcohols with a single −OH group are most suitable for obtaining small nanoparticles. With the increase in the −OH group, the size of synthesized ZnO nanoparticles gradually increases.

Different operation parameters have a significant impact on the formation mechanism of nano ZnO, and more study is still required in dealing with the nanoparticle formation mechanism. Studying the effects of different parameters on the supercritical hydrothermal synthesis of nano ZnO particles facilitates optimizing synthesis conditions, exploring reaction mechanisms, improving synthesis efficiency, and obtaining the required nanoparticles with specific morphology and properties. Furthermore, the solvent effect in the hydrothermal synthesis plays a significant role in the formation of zinc oxide nanorods. The solvent effect refers to the influence of the solvent on reaction rates, equilibrium constants, and reaction mechanisms. During the reaction process with the addition of the solvent, the solvent affects the growth rate of zinc oxide particles by modifying the solubility of intermediate products, the polarity and viscosity of the reaction medium, and the interactions between the reaction medium and reactants.

In this study, nano ZnO powders were prepared using supercritical hydrothermal synthesis technology. The effects of the reaction temperature, pressure, reaction time, precursor concentration, and ethanol content on the particle size, crystallinity, and morphology of the products were studied using the controlled variable method. The influence of reaction parameters on the ZnO formation mechanism, the best synthesis conditions, and the mechanism of adding organic solvent to prevent particle aggregation were determined.

## 2. Materials and Methods

### 2.1. Reagent Materials

Zinc acetate (Zn(CH_3_COO)_2_·2H_2_O, mass fraction >99.0%, Sinopharm, Beijing, China), sodium hydroxide (NaOH, mass fraction >90.0%, Sinopharm, Beijing, China), ethanol (C_2_H_5_OH, mass fraction >75.0%, Sinopharm, Beijing, China), and deionized water were used.

### 2.2. Experimental Procedure

Firstly, the 1 mol/L Zn(CH_3_COO)_2_·2H_2_O and NaOH solutions were prepared separately. For each experiment, a certain amount of the solution was taken, with a molar ratio of Zn(CH_3_COO)_2_·2H_2_O/NaOH of 1:2. The solutions were mixed evenly through ultrasonic mixing for 15 min in a microreactor (as shown in Figure 1, made of stainless steel 316, with an inner diameter and length of 14 and 80.0 mm, respectively). The microreactor was then properly sealed with a screw-sealed cap. The sealed reactor was placed in a tubular furnace that was heated to a specified temperature (i.e., reaction temperature). The self-generated pressure inside the microreactor reached the desired value (i.e., reaction pressure). After reacting for the predetermined time, the microreactor was quickly removed from the tubular furnace for shock cooling in a water bucket. The reaction products were then collected into a centrifuge tube, and the products were centrifuged to obtain the upper suspended liquid and the lower precipitate. The precipitate product was washed with ethanol and then centrifuged at least three times. The obtained sample was prepared for detection after being placed on the glass slide and vacuum-dried for over 12 h.

### 2.3. Material Characterization and Analysis Methods

X-ray diffraction (XRD) instrument with Cu-Kα was used for checking the crystallinity and phase of obtained powders. The phase composition and purity of the products were analyzed using Jade 9 software, and the crystallite size of ZnO was calculated using Scherrer’s Equation (1):(1)$$Size=\frac{K\lambda }{FW\left(S\right)*COS\left(\theta \right)},$$
where *Size* represents the crystallite size (nm), *K* is a constant typically set as *K* = 1, *λ* is the wavelength of X-ray (nm), *FW*(*S*) is the sample broadening (Rad), and *θ* is the diffraction angle (Rad).

The particle size and morphology of the sample were detected using field emission scanning electron microscopy (FESEM), whose acceleration voltage was 200 V~30 kV, and magnification was 16 X~1270 kX. To accurately study the particle size and its distribution, the Nano Measure 1.2 software tool was used to measure the particles from the FESEM micrographs by randomly selecting 100 measurement points. Furthermore, the measured particle sizes from the 100 points can be statistically plotted to create a size distribution bar graph, allowing for visual observation of the size distribution. By comparing the calculated crystallite size with the measured particle size, some conclusions can be obtained.

### 2.4. Experimental Conditions for Each Group

This study explored the effects of the reaction temperature, pressure, time, reactant concentration, and the molar ratio of precursor to ethanol on the size, crystallinity, and morphology of synthesized zinc oxide particles. Based on the control variates method, five sets of experiments were designed, as shown in Table 1, Table 2, Table 3, Table 4 and Table 5.

## 3. Results and Discussion

### 3.1. Effect of Reaction Temperature

The fixed reaction pressure was 26 MPa, the reaction time was 10 min, the concentration of Zn(CH_3_COO)_2_·2H_2_O solution was 0.3 mol/L, and no ethanol was added. The effect of the reaction temperature on the particle size, morphology, and crystal structure was studied. The XRD patterns of the products are shown in Figure 2. By comparing with the standard cards, the peak positions located at 31.8°, 34.4°, 36.3°, 47.5°, 56.6°, 62.8°, and 67.9° are consistent with the hexagonal wurtzite (JCPDS PDF#75-0576). The diffraction peaks are sharp, and no other impurity peaks are observed, indicating that the product is a well-crystallized and high-purity ZnO powder.

SEM images of the synthesized nano ZnO under different temperatures ranging from 300 °C to 500 °C are shown in Figure 3. It can be observed that the morphology of the ZnO product changes with increasing temperature. At a reaction temperature of 300 °C, the particles appear to be clustered into flower-like clusters and fragments (Figure 3a). As the reaction temperature increases to 350 °C, the product gradually transforms into a flower-like structure composed of conical nanoneedles (Figure 3b). When heated to 400 °C, the morphology begins to transform into hexagonal prismatic aggregates, but there are still flower clusters and flaky particles (Figure 3c). Subsequently, at 450 °C, the ZnO particles observed under FESEM exhibit a significant reduction in particle size and show a regular polyhedral morphology with a clear boundary (Figure 3d,e).

The change in ZnO morphology is caused by the change in ZnO solubility in supercritical water. The solubility of solutes in supercritical hydrothermal conditions mainly depends on the density of water. In the subcritical region (300–350 °C), since water has not reached the critical point, its density is higher than that of water in the supercritical state, ZnO solubility in the reaction medium is higher, and supersaturation of zinc oxide is lower, leading to a slow crystal growth rate [22]. So, the synthesized nanoparticles are irregular and uneven, which is another piece of evidence of low crystallinity. However, as the temperature increases to the supercritical region (400–500 °C), lower ZnO solubility in supercritical water results in a higher supersaturation, promoting the crystal growth of the product. The synthesized particles gradually take on a polyhedral shape with clear boundaries and tend to become uniform. This change indicates that a supercritical temperature is conducive to the formation of high-crystallinity crystals.

Since the morphology of ZnO generated at 300 and 350 °C is irregular, it is difficult to measure its particle size, so only the size distribution diagram of the product at 400–500 °C is shown here, as shown in Figure 4. It is the measured size distribution by randomly selecting 100 ZnO particles from the FESEM images at different temperatures. Through observation and analysis, it was found that when the reaction temperature reaches 450 °C (Figure 4b), the particle size of ZnO is the smallest, with an average size of 63.96 nm. As the temperature increases from 400 °C, the size of ZnO particles gradually decreases within a certain range. However, when the temperature reaches 500 °C, the size suddenly increases. Figure 5 shows the size variation of ZnO grains synthesized at different temperatures calculated by Equation (1). It is found that the variation trend is the same as that in Figure 4: when the reaction temperature is 400 °C, the ZnO grain size is the smallest, which is 33.9 nm. When the reaction temperature reaches 500 °C, the crystallite size increases to 50.1 nm. Within a certain range (300–400 °C), as the reaction temperature increases, the crystallite size of ZnO decreases. On the one hand, this is because the dielectric constant of water in the supercritical region is extremely low, resulting in low solubility of metal oxides, thereby improving the formation rate of ZnO crystal nuclei. The increase in the nucleation rate leads to a decrease in average particle size and crystallite size. On the other hand, according to the Born equation, the reaction rate is inversely proportional to the dielectric constant [10]. Therefore, the extremely low dielectric constant leads to faster hydrolysis, dehydration, nucleation, and growth rates compared to the subcritical region, resulting in a gradual decrease in the average particle size, which also improves the yield of ZnO particles to a certain extent. After the temperature reaches 400 °C, the Ostwald ripening and collision probability between ZnO particles increase and have an increasing influence on particle growth, leading to an increase in particle aggregation and, subsequently, an increase in the crystallite size and particle size [23,24].

### 3.2. Effect of Reaction Pressure

Under experimental conditions fixed at 450 °C, 10 min, a Zn(CH_3_COO)_2_·2H_2_O solution concentration of 0.3 mol/L, and no ethanol addition, reaction pressure variation from 22 to 30 MPa was carried out to investigate the effect of pressure on the preparation of ZnO particles. The XRD spectra of the obtained products can be found in the Appendix A (Appendix A). After matching, it was found that the crystal plane height of the samples was highly consistent with the hexagonal wurtzite structure, indicating that the prepared product was zinc oxide nanoparticles with high purity.

Figure 6 shows the FESEM spectra of synthesized nano ZnO under different pressure conditions. It can be observed that at 22 MPa, the morphology of the synthesized ZnO crystals (Figure 6a) is an irregular polygonal aggregate with varying particle sizes. As the pressure increases, ZnO particles gradually become more homogeneous and dispersed than before (Figure 6b–d). This is because the increase in pressure from the sub-region to the super-region in the hydrothermal synthesis process will increase the reaction rate, promote condensed matter nucleation in the reaction system, and make the substances in the solution more uniform in diffusion and migration, thus promoting the nucleation and growth of particles [24,25,26]. With the pressure increasing from 24 MPa to 28 MPa, the synthesized ZnO crystals maintain a hexagonal prism shape, but the boundaries are clearer and more evenly distributed, with the average particle size increasing slightly from 70.36 to 84.08 nm (as shown in Figure 7). Continuing to increase the pressure to 30 MPa, it is found that the ZnO crystal size becomes larger, the distribution becomes uneven, and particle aggregation can be found in the FESEM photograph (Figure 6e). These phenomenons are attributed to the slight increase in water density and dielectric constant in the supercritical state [27,28]. A higher reaction medium density and dielectric constant weaken the advantages of supercritical water, adverse to crystal nucleation, resulting in larger particle size and even ununiform particles [29,30,31]. The variation in crystallite size (as shown in Appendix A of the Appendix A) from 22 MPa to 30 MPa exhibits a similar trend to the average particle size, which is other proof of our deduction.

### 3.3. Effect of Reactant Concentration

Under experimental conditions of 450 °C, 26 MPa, 10 min, and no ethanol addition, different Zn(CH_3_COO)_2_·2H_2_O concentrations (0.1 mol/L, 0.2 mol/L, 0.3 mol/L, 0.4 mol/L, and 0.5 mol/L) were investigated to explore the effect of the reactant concentration on the preparation of ZnO particles. The prepared particles were characterized by XRD and FESEM, as shown in Appendix A (which can be found in the Appendix A) and Figure 8, respectively. XRD spectra graphs confirm the production of ZnO crystals, and no impurity can be found.

Figure 8 and Figure 9 show FESEM images and the particle size distribution of ZnO synthesized with different concentrations of Zn(CH_3_COO)_2_·2H_2_O. As can be seen in the figure, the particle size and shape of ZnO nanoparticles are also affected by the precursor concentration; with the increase in the precursor concentration, the particle size tends to decrease first and then increase. When the precursor concentration increases from 0.1 mol/L to 0.3 mol/L, the morphology of ZnO particles gradually changes from an irregular polyhedron to uniformly distributed hexagonal prisms, and the particle size decreases from 83.48 nm to 63.96 nm (Figure 8a–c). However, when the precursor concentration exceeds 0.3 mol/L and increases, the particle morphology returns to an irregular state, and the particle size also increases to 79.94 nm (Figure 8e). This appears to point to the existence of an optimum concentration for producing the smallest particle size of ZnO nanoparticles.

According to Sue et al. [32], when the initial reactant concentration is much higher than the solubility of the metal oxide, a decrease in the initial reactant concentration usually results in the formation of smaller particles. When the initial concentration of precursors was reduced and approached the solubility limit of ZnO, the dissolution and precipitation of ZnO were accelerated. At the same time, according to the classical LaMer theory [33], the initial parent ion concentration determines the supersaturation of the solution, and the effect of supersaturation on the nucleation rate is that supersaturation provides the driving force for nucleation. Nucleation is the transformation process of the supercritical precursor to the solid phase, which requires driving force. Only when the concentration of parent ions in the solution has a certain supersaturation can the formation of a new phase be driven. The higher the supersaturation, the greater the phase transition driving force and, therefore, the higher the nucleation rate. The increasing concentration of the precursor increases the driving force for nucleation, which takes on an explosive state and rapidly forms uniform particles. This is the cause of uniform particle morphology at 0.1–0.3 mol/L. However, with a further increase in precursor concentration, the deposition solubility increases, and the particle size of nano-ZnO expands. Therefore, when the concentration is greater than 0.3 mol/L, the particle size of nano-ZnO increases gradually [26,34,35,36,37]. The particle size change from 0.1 to 0.5 mol/L (as shown in Appendix A in the Appendix A) is another basis for our conclusion.

### 3.4. Effect of Reaction Time

Under experimental conditions of a temperature of 450 °C, pressure of 26 MPa, Zn(CH_3_COO)_2_·2H_2_O solution concentration of 0.3 mol/L, and 0 ethanol addition, reaction times of 5 min, 7.5 min, 10 min, 12.5 min, and 15 min were investigated to determine the effect of the reaction time on the preparation of ZnO particles. The XRD spectra of obtained products can be found in the Appendix A (Appendix A). After matching, it was found that the crystal plane height of the samples was highly consistent with the hexagonal wurtzite structure.

Figure 10 shows FESEM images of ZnO obtained at different reaction times. It can be seen in Figure 10 that when the reaction time is 5 and 10 min, the morphology of ZnO particles generated is mainly foliate-like and needle-like and gradually changes to the regular hexagonal prism shape. When the reaction time is 5 min (Figure 10a), nano-ZnO is foliated, which is because the reaction time is too short, the reaction is not sufficient, and the growth process of ZnO crystals is not complete. Some factors cause the reaction conditions to not reach the ideal state due to the short heating time of the reactor and uneven heating. As the reaction time increases (Figure 10b), leaf-shaped ZnO gradually changes into needle-shaped particles. This is because the increase in the reaction time gives ZnO particles sufficient time to grow. At this time, both radial and axial ZnO particles are growing, so most of them are needle-shaped particles. With the further extension of the reaction time (10 min, Figure 10c), ZnO nanoneedles have sufficient time for growth, and their axial growth rate and radial growth rate tend to balance. Finally, a regular uniform distribution of hexagonal prismatic particles is formed [22,24,38]. At the same time, the size distribution of ZnO particles prepared with a reaction time ranging from 10 to 15 min (Figure 11) also reflects this feature. When the reaction time increases from 10 min to 15 min, the ZnO particle size gradually increases, the agglomeration phenomenon intensifies, and the morphology becomes an irregular polyhedron, which can be explained by the Ostwald effect [39,40]. Moreover, the crystallite size changes of ZnO synthesized at different times (as shown in Appendix A in the Appendix A) calculated by Equation (1) also have the same trend.

### 3.5. Effect of the Amount of Ethanol Addition 

Under experimental conditions of a temperature of 450 °C, pressure of 26 MPa, Zn(CH_3_COO)_2_·2H_2_O solution concentration of 0.3 mol/L, and reaction time of 10 min, the molar ratios of precursor to ethanol were selected as 2:1, 1:1, 1:2, 1:3, and 1:4, respectively, to investigate the effect of ethanol addition on the preparation of ZnO particles. The XRD spectra of the obtained products can be found in the Appendix A (Appendix A). After matching, it was found that the crystal plane height of the samples was highly consistent with the hexagonal wurtzite structure.

Figure 12 and Figure 13 show the FESEM diagram and particle size distribution diagram of ZnO synthesized at different ethanol concentrations, respectively. It can be seen that the addition of ethanol has little effect on the ZnO grain size. With the increase in the ratio of the precursor to ethanol, the ZnO grain size fluctuates around 80 nm. The morphology and dispersion of the particles are greatly affected, and the particles gradually change from irregular polyhedrons and rod-like structures to uniformly dispersed spherical particles. The particle size changes due to the chemical bonding between ethanol and ZnO, which occurs through adsorption and chemical interactions. This bonding can occur in two ways: (1) covalent bonding between the positively charged ZnO surface and the dissociated part of R–OH; (2) bonding of hydroxyl groups on the ZnO surface with the hydroxyl groups of R–OH. The secure attachment of R–OH on the ZnO surface leads to a reduction in particle size [41]. The “dissolution and crystallization process” theory has been considered a basic mechanism to describe the hydrothermal process, and in the dissolution–crystallization process, the physical properties of the solvent, such as permittivity and surface tension, also affect the process [41,42]. When there is no ethanol in the solvent or the ethanol content is low, the surface tension of the solvent is high, and the hydroxide precipitate will aggregate, resulting in a longer time for dehydration to ZnO powder. If ethanol is added to the solvent, the surface tension of the solvent will be reduced, resulting in the hydroxide precipitate being wrapped faster by the surrounding solvent, thus exhibiting better dispersibility and easier dehydration to form ZnO powder. In addition, the introduction of ethanol in water can reduce the dielectric constant of the solvent, thereby reducing the solubility of ZnO and increasing the saturation of ZnO, thus increasing the nucleation rate [43,44].

Figure 14 shows the dispersion test of ZnO synthesized in three different media under the conditions of 450 °C, 26 MPa, precursor concentration of 0.3 mol/L, reaction time of 5 min, and molar ratio of precursor to ethanol of 1:3. It can be observed in the figure that the ZnO nanoparticles synthesized under this condition form a uniform suspension in the three media. This indicates that the synthesized ZnO nanoparticles have a good dispersion in different media.

Figure 15 shows FT-IR (Fourier Transform infrared spectroscopy) of ZnO NPs synthesized with the addition of organic solvents of different molar ratios. It can be seen in the figure that the FT-IR curves of ZnO NPs synthesized by adding different moles of ethanol are very similar. By comparing the infrared spectra with those of ordinary ZnO, it was found that the characteristic peaks of the product were basically consistent with those of ordinary ZnO. Absorption peaks in the range of 3500 to 3300 cm^−1^ can be found in the figure, which correspond to stretching vibrations of hydroxyl (OH) groups. There are several weak absorption peaks in the range of 1700~1400 cm^−1^, which may be the peaks of molecular vibration and bending vibration of hydroxyl coordination compounds in ZnO. The absorption peak between 700 and 400 cm^−1^, which is the characteristic absorption peak of ZnO, corresponds to the bending vibration of Zn-O bonds in ZnO. It can be seen that the intensity of the absorption peaks of ZnO NPs synthesized by adding different molar ratios of ethanol is different. This is because the particle size of ZnO NPs synthesized by adding ethanol is small, and the proportion of surface atoms is large, so the lattice vibration under infrared irradiation is different from that of ordinary synthesized ZnO NPs [45,46,47].

## 4. Conclusions and Prospects

ZnO nanoparticles with high dispersion and a small particle size were prepared using the supercritical hydrothermal synthesis technique. By adjusting the reaction temperature, pressure, precursor concentration, reaction time, and the ratio of precursor to ethanol, ZnO nanoparticles with different shapes and particle sizes were synthesized. The influence of reaction conditions on the product was studied by analyzing its morphology, particle size, and dispersion degree.

The results show that the morphology and particle size of nano-ZnO are greatly affected by the reaction conditions. The particle size of ZnO decreases with an increase in temperature and pressure within an appropriate range due to the extremely low dielectric constant and density of water in a supercritical state. These physical properties result in a sharp decrease in solubility of the product in supercritical water, leading to rapid formation.

The influence of the precursor concentration and reaction time on the particle size is mainly determined by nucleation and grain growth. At low precursor concentrations, nucleation is slowed down due to a lack of driving force, which contributes to relatively large particles being produced. If the reaction time is too short, there will be insufficient reaction between precursors, resulting in smaller particles not forming properly.

Additionally, adding ethanol plays a key role in improving both size and morphology by establishing chemical bonds with ZnO while inhibiting agglomeration. The morphology and dispersion of ZnO synthesized with each proportion of ethanol are better than those without ethanol, indicating that an organic solvent may be better than supercritical water in the synthesis of nanoparticles. Therefore, the optimum synthesis conditions of ZnO are 450 °C, 26 MPa, 0.3 mol/L, 10 min, and a molar ratio of precursor to ethanol of 1:3.

The supercritical hydrothermal synthesis of ZnO nanoparticles is an exciting new technique. The reaction conditions and parameters have a great influence on the size and morphology of ZnO nanoparticles. Therefore, the advantage of supercritical hydrothermal synthesis technology compared with other technologies is that the size, morphology, and dispersion of its products can be easily controlled by changing the conditions and parameters or adding surface modifiers during the reaction process.

However, although we have conducted some research on this technology, there are still areas where these studies need to be improved: First, most current research has focused on the effect of a single factor. In fact, the influence of parameters is not unilateral, and there may be coupling effects between various factors. Therefore, it is very important to optimize multiple parameters of the supercritical hydrothermal process. The response surface method is an excellent method for process optimization and has been used to study the influence of factors in the synthesis of other nanomaterials. Therefore, in the study of the supercritical hydrothermal synthesis of ZnO nanoparticles, the same optimization of each reaction parameter is needed to achieve the best process conditions.

In addition, incorporating modifiers into supercritical hydrothermal synthesis is a fascinating technique. The addition of a modifier can not only influence the particle size and morphology of nano-ZnO but also significantly enhance its performance. However, there is limited research on ZnO synthesis by adding a modifier in supercritical water. Therefore, the synthesis of ZnO nanoparticles with optimal properties in various fields by incorporating modifiers in the supercritical hydrothermal process will be a promising research direction.

## Figures and Tables

**Figure 1 nanomaterials-14-00844-f001:**
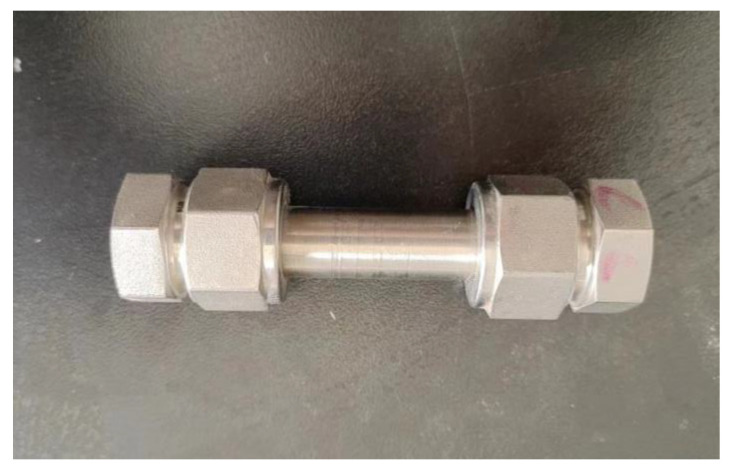
Microreactor schematic diagram.

**Figure 2 nanomaterials-14-00844-f002:**
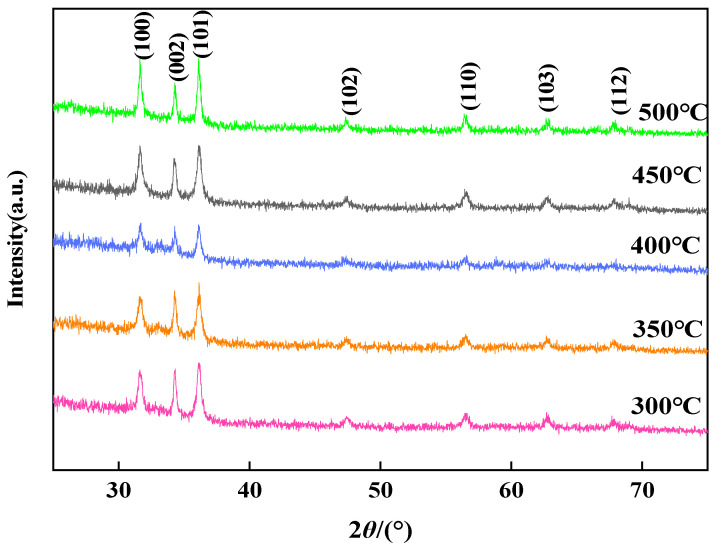
XRD spectra of nano ZnO synthesized at different temperatures.

**Figure 3 nanomaterials-14-00844-f003:**
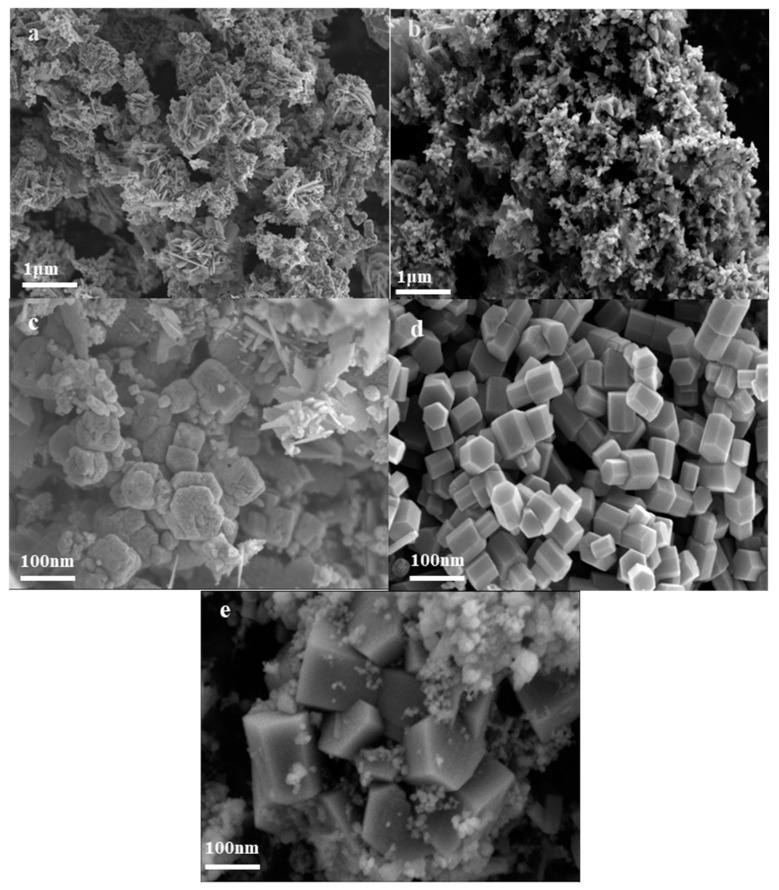
FESEM images of synthesized nano ZnO under different temperature conditions: (**a**) 300 °C; (**b**) 350 °C; (**c**) 400 °C; (**d**) 450 °C; (**e**) 500 °C.

**Figure 4 nanomaterials-14-00844-f004:**
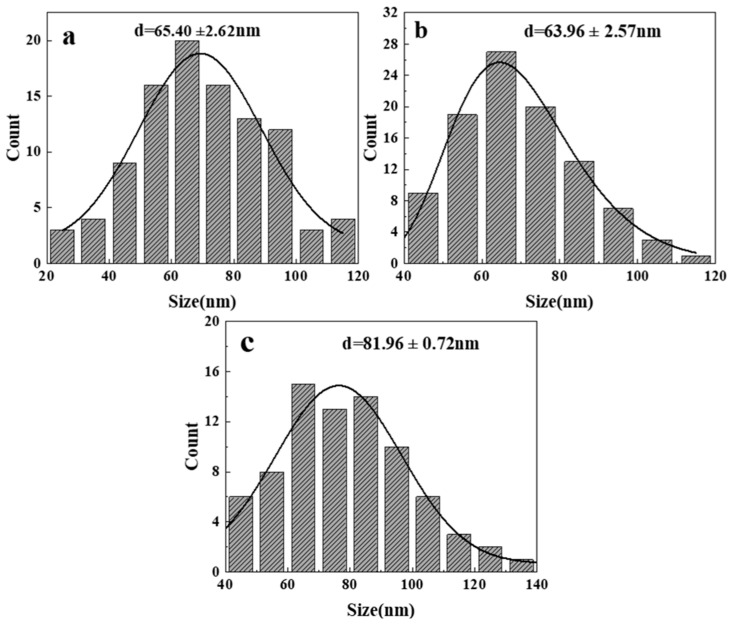
Size distribution of synthesized ZnO at different temperatures from FESEM images: (**a**) 400 °C; (**b**) 450 °C; (**c**) 500 °C.

**Figure 5 nanomaterials-14-00844-f005:**
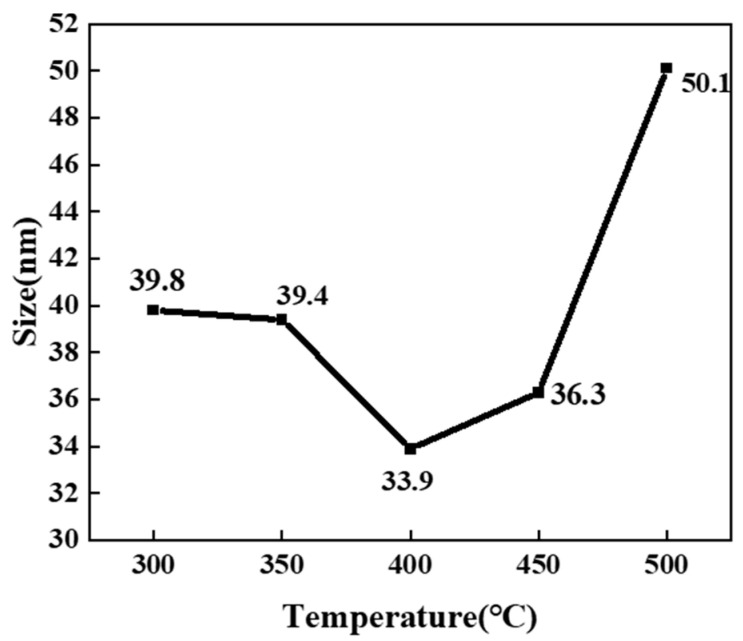
Variation in crystallite size of ZnO synthesized from 300 °C to 500 °C.

**Figure 6 nanomaterials-14-00844-f006:**
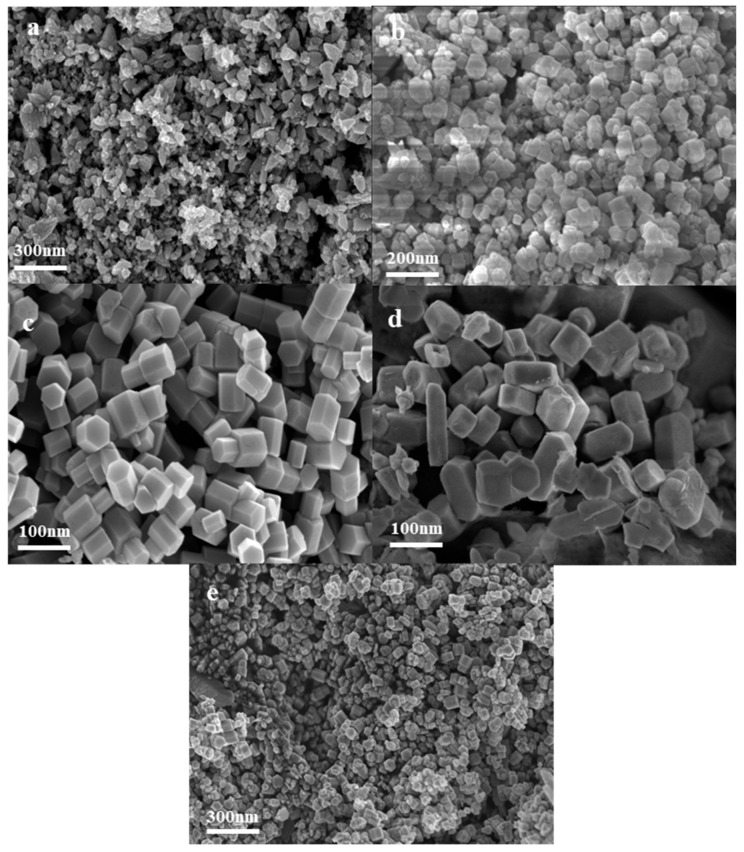
FESEM images of synthesized nano ZnO under different pressure conditions: (**a**) 22 MPa; (**b**) 24 MPa; (**c**) 26 MPa; (**d**) 28 MPa; (**e**) 30 MPa.

**Figure 7 nanomaterials-14-00844-f007:**
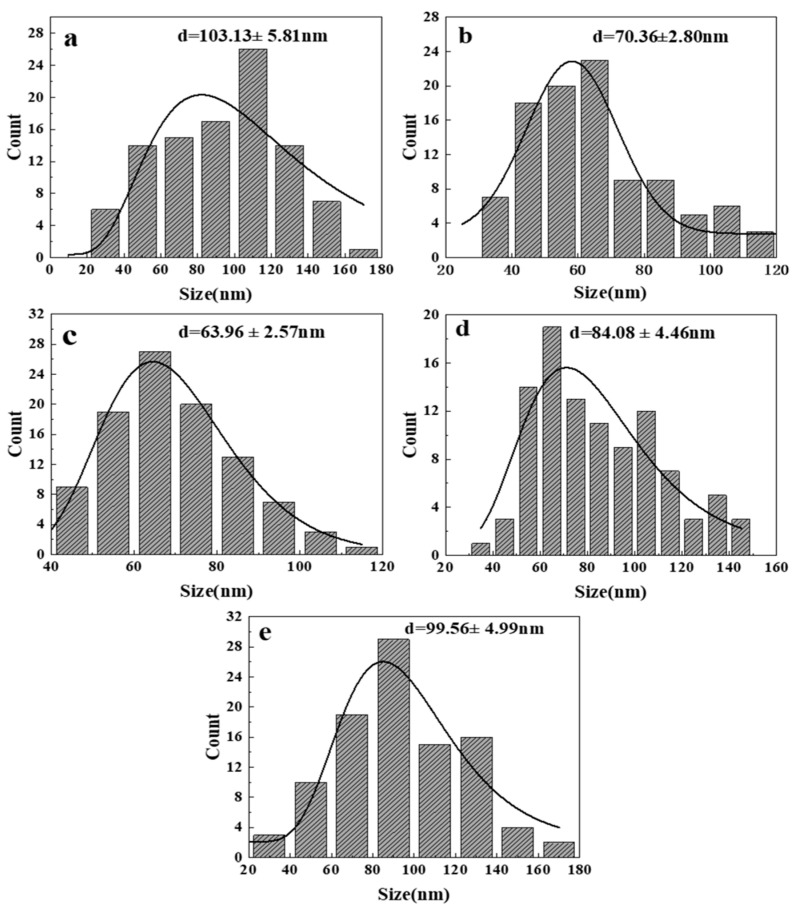
Size distribution of synthesized ZnO at different pressures from FESEM images: (**a**) 22 MPa; (**b**) 24 MPa; (**c**) 26 MPa; (**d**) 28 MPa; (**e**) 30 MPa.

**Figure 8 nanomaterials-14-00844-f008:**
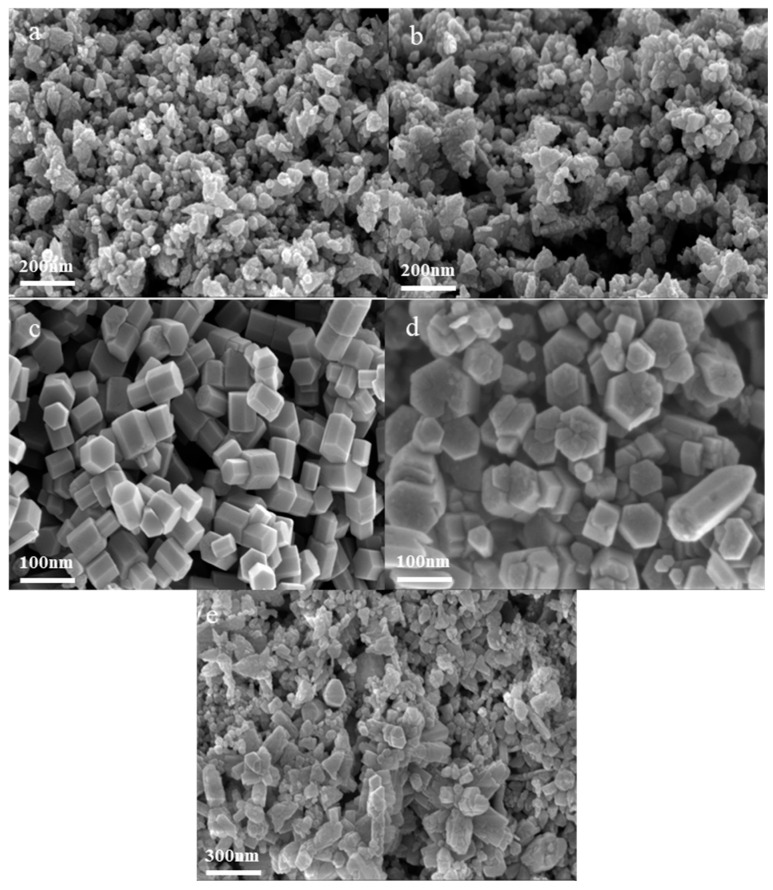
FESEM images of synthesized nano ZnO under different concentration conditions: (**a**) 0.1 mol/L; (**b**) 0.2 mol/L; (**c**) 0.3 mol/L; (**d**) 0.4 mol/L; (**e**) 0.5 mol/L.

**Figure 9 nanomaterials-14-00844-f009:**
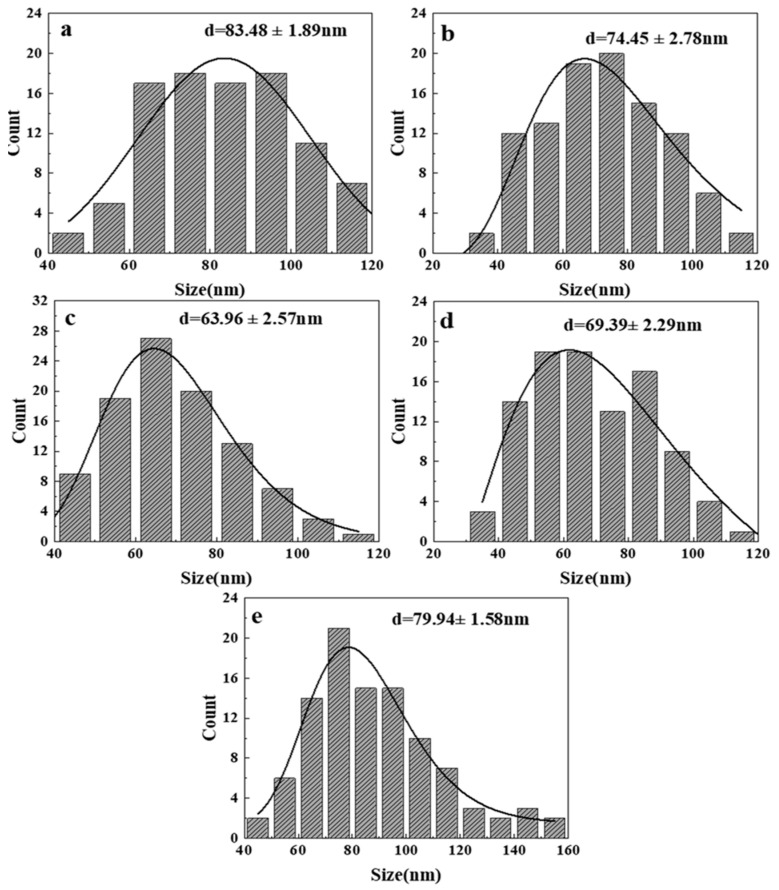
Size distribution of synthesized ZnO at different concentrations from FESEM images: (**a**) 0.1 mol/L; (**b**) 0.2 mol/L; (**c**) 0.3 mol/L; (**d**) 0.4 mol/L; (**e**) 0.5 mol/L.

**Figure 10 nanomaterials-14-00844-f010:**
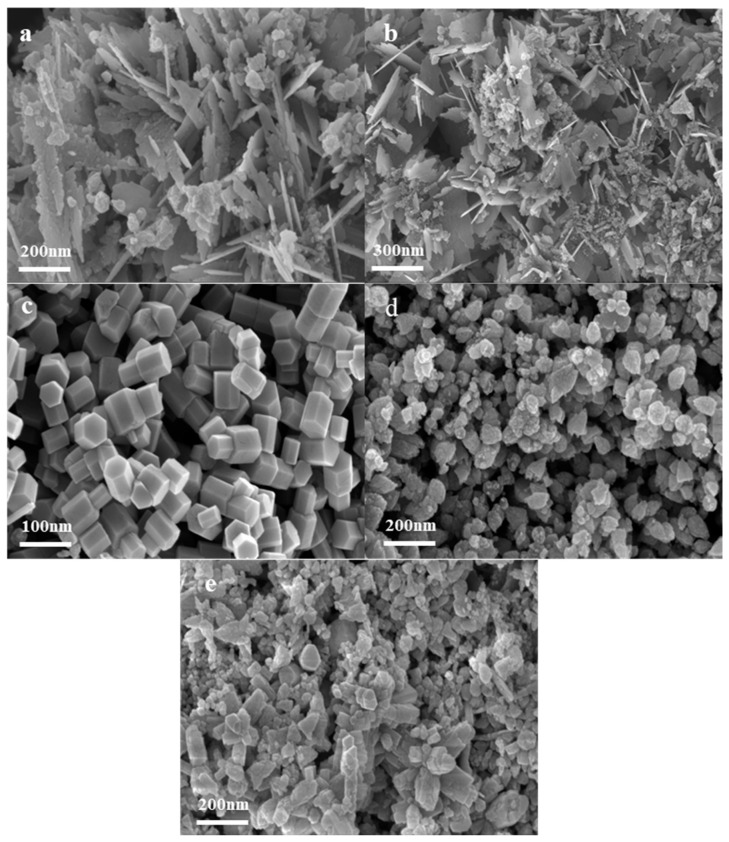
FESEM images of synthesized nano ZnO under different reaction times: (**a**) 5 min; (**b**) 7.5 min; (**c**) 10 min; (**d**) 12.5 min; (**e**) 15 min.

**Figure 11 nanomaterials-14-00844-f011:**
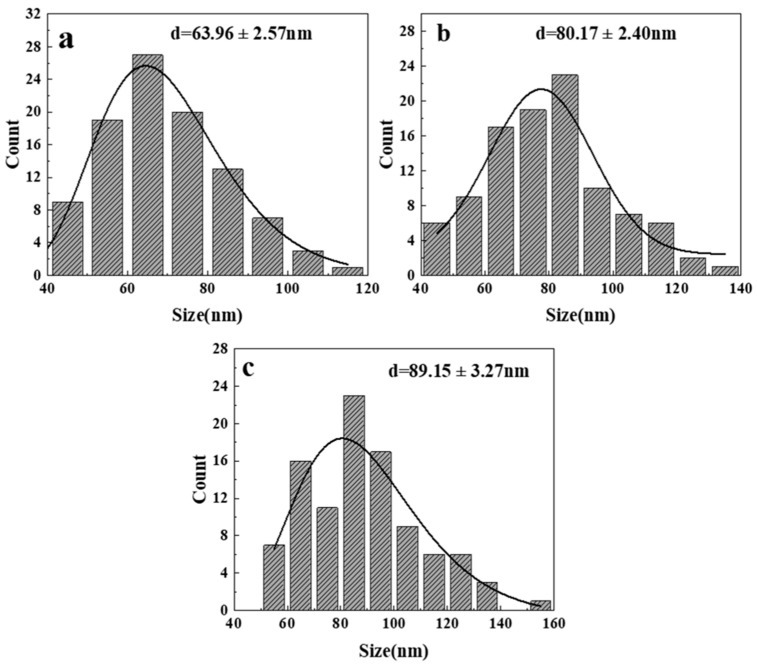
Size distribution of synthesized ZnO at different reaction times from FESEM images: (**a**) 10 min; (**b**) 12.5 min; (**c**) 15 min.

**Figure 12 nanomaterials-14-00844-f012:**
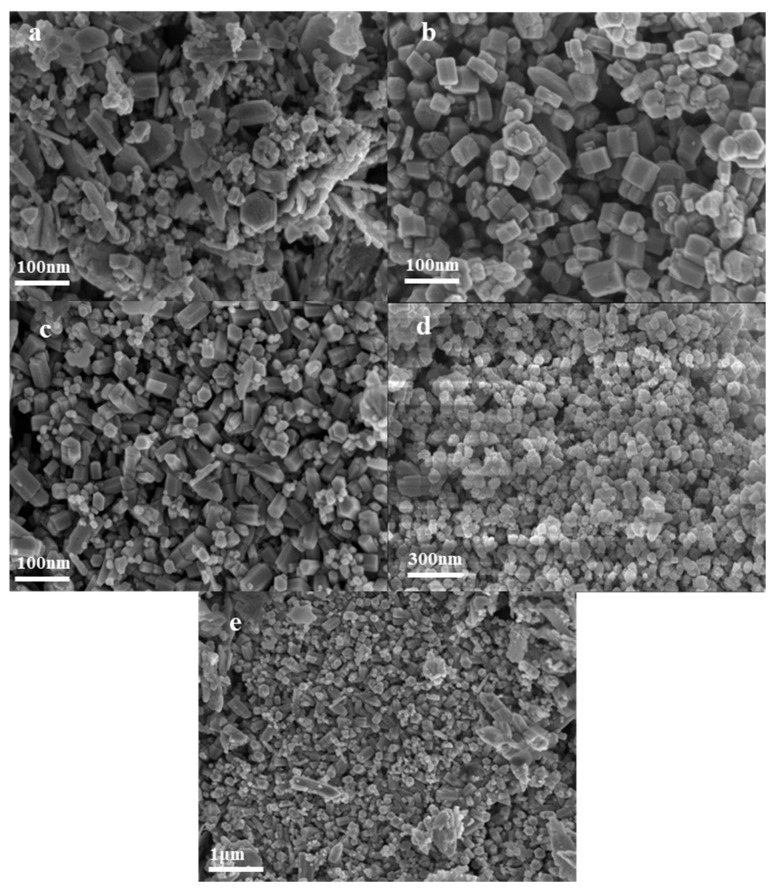
FESEM images of synthesized nano ZnO under different amounts of ethanol addition: (**a**) 2:1; (**b**) 1:1; (**c**) 1:2; (**d**) 1:3; (**e**) 1:4.

**Figure 13 nanomaterials-14-00844-f013:**
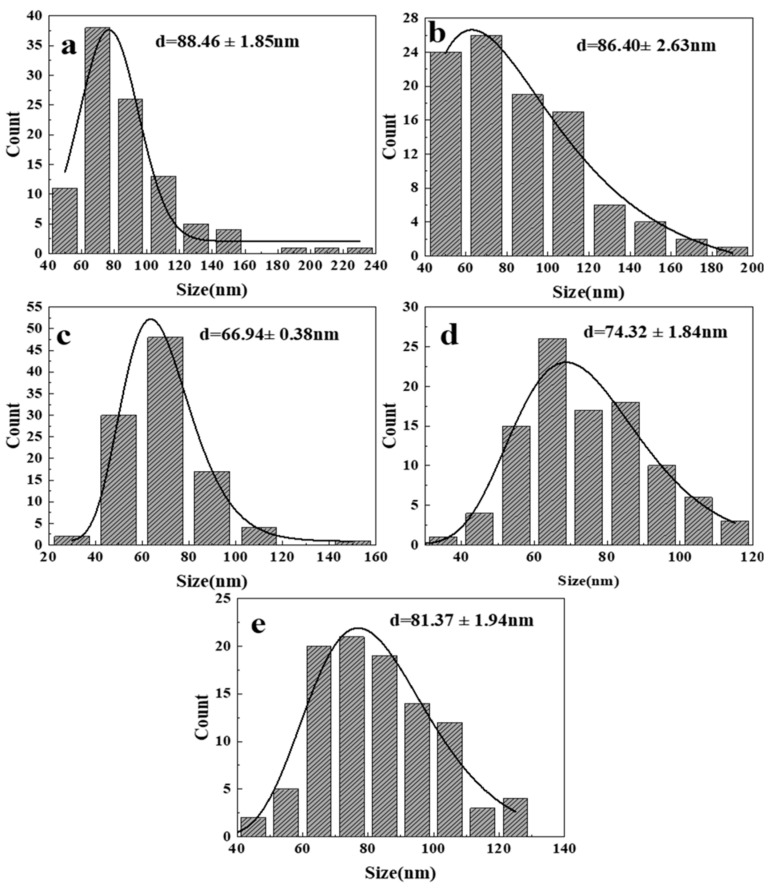
Size distribution of synthesized ZnO under different amounts of ethanol addition from FESEM images: (**a**) 2:1; (**b**) 1:1; (**c**) 1:2; (**d**) 1:3; (**e**) 1:4.

**Figure 14 nanomaterials-14-00844-f014:**
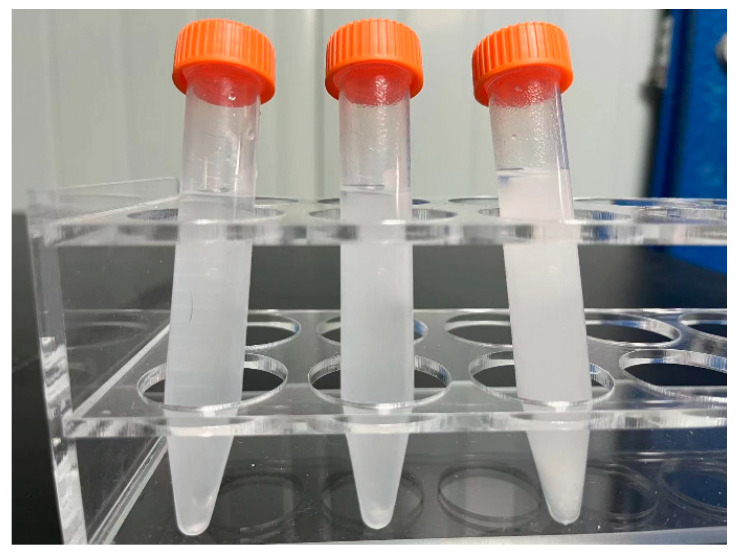
The dispersion of ZnO NPs in different media, from left to right: dispersed in water, ethanol, ethylene glycol.

**Figure 15 nanomaterials-14-00844-f015:**
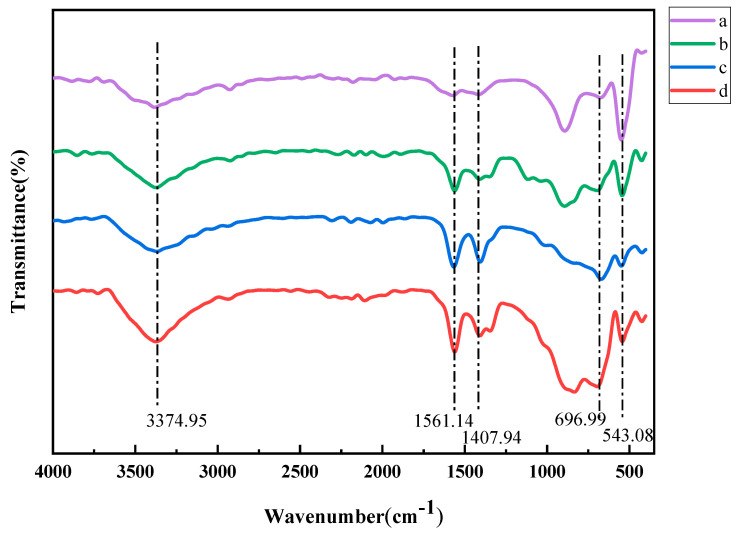
FT-IR of ZnO NPs determined using supercritical hydrothermal synthesis under a molar ratio of the precursor to ethanol as follows: (a) 1:0; (b) 1:1; (c) 1:2; (d) 1:3.

**Table 1 nanomaterials-14-00844-t001:** Reaction conditions of Experimental Group 1 without the addition of ethanol.

No.	Temperature (°C)	Pressure (MPa)	Time (min)	Precursor Concentration (mol/L)
1	300	26	10	0.3
2	350	26	10	0.3
3	400	26	10	0.3
4	450	26	10	0.3
5	500	26	10	0.3

**Table 2 nanomaterials-14-00844-t002:** Reaction conditions of Experimental Group 2 without the addition of ethanol.

No.	Temperature (°C)	Pressure (MPa)	Time (min)	Precursor Concentration (MPa)
1	450	22	10	0.3
2	450	24	10	0.3
3	450	26	10	0.3
4	450	28	10	0.3
5	450	30	10	0.3

**Table 3 nanomaterials-14-00844-t003:** Reaction conditions of Experimental Group 3 without the addition of ethanol.

No.	Temperature (°C)	Pressure (MPa)	Time (min)	Precursor Concentration (MPa)
1	450	26	10	0.1
2	450	26	10	0.2
3	450	26	10	0.3
4	450	26	10	0.4
5	450	26	10	0.5

**Table 4 nanomaterials-14-00844-t004:** Reaction conditions of Experimental Group 4 without the addition of ethanol.

No.	Temperature (°C)	Pressure (MPa)	Time (min)	Precursor Concentration (MPa)
1	450	26	5	0.1
2	450	26	7.5	0.2
3	450	26	10	0.3
4	450	26	12.5	0.4
5	450	26	15	0.5

**Table 5 nanomaterials-14-00844-t005:** Reaction conditions of Experimental Group 5 with the addition of ethanol.

No.	Temperature (°C)	Pressure (MPa)	Time (min)	Precursor Concentration (MPa)	Mole Ratio between Zn^2+^ and Ethanol
1	450	26	10	0.1	2:1
2	450	26	10	0.2	1:1
3	450	26	10	0.3	1:2
4	450	26	10	0.4	1:3
5	450	26	10	0.5	1:4

## Data Availability

Data is contained within the article.

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
