# Peer review of "Study on the Synthesis of Nano Zinc Oxide Particles under Supercritical Hydrothermal Conditions"

_nanomaterials, 2024, doi:10.3390/nano14100844_

Round 1

Reviewer 1 Report

Comments and Suggestions for Authors

The article "Study on the Synthesis of Nano Zinc Oxide Particles under Supercritical Hydrothermal Conditions" describes the influence of various synthesis parameters (temperature, concentration, pressure, time, additional solvent) on the size and morphology of ZnO nanoparticles. It is a valuable study that can be published after authors address the following problems:

Abstract should be checked and revised carefully by briefly introducing the work plan and key findings.

Abstracts should highlight the innovation of the article, as often abstract section is presented separately in search engines, it must be able to stand alone as an informative piece. Please add the precursor and the solvent information in Abstract.

In introduction a stronger recent literature survey is necessary, especially on previous literature reports on the influence of solvent (alcohols) on ZnO synthesis. This work is interesting and can be boosted further. Hence the following literature could prove this manuscript: doi: 10.3390/pharmaceutics14122842 and doi: 10.3390/ijms24065677.

How was the pressure of 26 MPa kept at fixed value (row 138) as long as authors varied the temperature and added ethanol (authors declare a self-generated pressure, therefore with no external adjustments)? Please refer also to the rows 205-206 “reaction pressure varied from 22 to 30 MPa”.

A schematic depiction of the working flux and reactor would improve the readability of the manuscript (please see doi: 10.3390/Ijms24021629).

It is not clear the way ethanol was added: a) add the number of ethanol moles (I suppose a calculated absolute ethanol volume) to the 0.3 M (zinc acetate in water) solution; in this case the volume of precursor solution was diminished to make room for the ethanol, and practically the concentration of zinc acetate decreased in the final mixture? As such there are two variables here, the ethanol amount and final concentration of the solution. b) made a 0.3 M solution with water and ethanol (in the mentioned molar ratio zinc acetate to ethanol), therefore keeping the concentration of precursor constant?

In XRD section, the last value 67.9o does not have corresponding Miller indices in figure 1, probably (112).

In my opinion the tabular information from Supplementary materials should be included in the main article.

Some minor typos must be corrected (e.g. Mpa from row 358).

The conclusion part does not highlight the salient findings and future perspective.

Author Response

Thank you for your critical comments and helpful suggestions. We have revised the manuscript accordingly. We have made a point-to-point reply to your questions and marked the modification location.

Reviewer 2 Report

Comments and Suggestions for Authors

Review comments on “Study on the Synthesis of Nano Zinc Oxide Particles under Supercritical Hydrothermal Conditions.” This manuscript is recommended for resubmission after the major revisions of the following comments:

1.      Line 106: "The self-generated pressure inside the microreactor will reach the desired value (i.e., reaction pressure)." How do the authors confirm the self-generated pressure quantitatively? This should be explained in detail.

2.      The labeling of the figures should be written properly. For instance, in Fig. 1, there should not be a "/" between Intensity and its unit.

3.      Line 203, section 3.2: Clarify how the authors varied the pressure. Some of the SEM images (for instance, Fig. 2d and 5c) are exactly the same. Do the authors mean that at the provided reaction conditions, the zinc oxide particles are the same and there is no effect of changing conditions on particle morphology? The authors need to provide the SEM images with their original scale bar.

4.      Why did the authors only use ethanol? What happens if a modifier is also used with ethanol?

5.      Maintain uniformity in referencing. For example, please refer to references 4 and 5 and remove literature that is not available online.

6.      In section 3.2, the pressure is being varied from 22 to 30 MPa, but lacks explanation on how the authors varied the pressure. The process for pressure variation is not well-described and is concerning. Therefore, authors should consider scientific means for varying the pressure that can be replicated.

7.      Line 208: What is crystal plane height? It should be replaced with "crystal plane intensity". The higher the intensity of any plane, the more the particular facet dominates in the crystal.

8.      Regarding the reaction time, what is the scientific background to choose a time interval of 2.5 minutes? Please describe it in detail or substantiate with citation.

9.      The study has provided XRD peaks but has not described changes in structure based on crystal facets in detail, which is important for this study.

10.   Scanning electron microscopy and XRD alone are insufficient to characterize the materials and support the claims. Therefore, authors are recommended to perform some basic characterization techniques (FTIR, XPS, TGA, UV-Vis spectroscopy, etc.) to provide details on the as-synthesized materials.

11.   Checking the dispersibility of the synthesized particles is an important factor. Please check it in different media such as water, EtOH, and Glycol, etc.

Author Response

(The authors gave the same response as above.)

Reviewer 3 Report

Comments and Suggestions for Authors

The manuscript, submitted by Panpan Sun and coworkers, reports on a study of hydrothermal synthesis of nano – ZnO nanoparticles. The research area is current, important and it fits well into the topics covered by Nanomaterials.

However, after thorough reading, I found several shortcomings which should be resolved before considering the publication of the paper.

Abstract: in Line 10, the authors write ‘In this paper, …’. However, right now,  it is still a submitted manuscript. I suggest ‘In this study, …’. The same phrase is used several times further on through the text.

Abstract, line 12: While there is no doubt that ethanol is both organic and green, the use of this phrases in the Abstract seems a bit populistic and more appropriate for mass media than for a serious scientific journal.  A plethora of scientific papers has been published throughout decades, reporting the use of ethanol either as a solvent, or as a reactant, without using ‘green organic’.

Abtract, lines 13 – 18: The sentence is too long, difficult to follow and should be split in at least two shorter sentences.

Introduction, lines 25 – 29: The refences, which are meant to support the claims of the authors regarding the significance and usefulness of ZnO nanoparticles, are inappropriate. (2) – (4) are more than 20 years old while (1) is a master’s thesis from 2004 (!). More recent papers, preferably from high – ranked journals should be provided as references.

Introduction, line 55: the same remark as above: The authors use the phrase ‘At present’, which is followed by two master’s theses from 2011 and 2015. Please provide more recent references from scientific journals.

Materials and methods, Paragraph 2.2: This paragraph seems to have been written by another author than the Introduction and is written more like an instruction for laboratory work (prepare … mix … seal) than a description of the performed experiments. It should be written in past tense, e.g., ‘1 M solution was prepared’.

Materials and methods, Paragraph 2.2, line 09: What is meant by ‘The microreactor collects … ‘? Please explain or rephrase.

Results and Discussion, Paragraph 3.1., line 41: ‘The XRD pattern is shown … ‘ should read ‘The XRD patterns are shown … ‘, since there are 5 of them.

FESEM figures (Figs. 2 – 7), general remarks: It is sometimes difficult to compare the pictures and thus follow the text due to different magnifications used inside the same figure (for example, let us look at Figure 5: the bar corresponds to 300 nm, 200 nm, 100 nm, 100 nm, and again 300 nm. Did the authors consider using the same magnification at least inside one figure?

Conclusions, 1st paragraph: The whole paragraph should be completely rewritten. It is full of typos (e.g., missing blank spaces before brackets and between the number and degrees Celsius), and grammatical errors (the present study focuses on, ZnO particles, … were prepared). In results and discussion, it can clearly be seen that ZnO nanoparticles of different sizes were prepared, yet in the 2nd line of Conclusions, the authors suddenly write how ‘particles … of 74.32nm’ were prepared – why pointing out only one size?  

Conclusions, 2nd paragraph: Again, the sentence is too long, difficult to follow and should be split into at least two shorter sentences.

Comments on the Quality of English Language

Not being an English speaker myself, I have the impression that the language is of decent quality but still needs a moderate editing. Some of the sentences are too long and should be split into shorter ones. There are major shortcomings in Materials and methods, Paragraph 2.2, and 1st paragraph of the Conclusions, as already stated in Comments and Suggestions for Authors.

Author Response

(The authors gave the same response as above.)

Round 2

Reviewer 1 Report

Comments and Suggestions for Authors

The authors have responded to my comments and have addressed all my concerns, substantially improving the manuscript, therefore, I suggest publishing the paper titled "Study on the Synthesis of Nano Zinc Oxide Particles under Supercritical Hydrothermal Conditions" in the current form.

Author Response

Thank you very much for your helpful suggestions and encouragement to us. Each of your comments has an important guiding significance for our work, so that we can continuously improve and perfect.

Reviewer 2 Report

Comments and Suggestions for Authors

The answer to Comment 10 is not sufficient. This is a critical issue in the characterization of the samples. I can't accept the manuscript in its current form.

Author Response

Many thanks to the distinguished reviewers for their comments and suggestions on our work. We are very sorry that our answer to comment 10 did not satisfy you. Therefore, we think it necessary to explain it to you again. XRD (X-ray Diffraction) is a widely recognized, widely used technique for fast, accurate, and efficient non-destructive testing of materials. As a means of characterizing the crystal structure and its change law, it is widely used in many fields such as materials, chemistry, biology, medicine, ceramics, metallurgy, minerals and so on.    Its main role in the field of material analysis is(1) phase characterization; (2) Determine the cell parameters; (3) crystal orientation analysis; (4) grain size calculation; (5) Phase quantitative calculation. In our study, we mainly used XRD to characterize the phase and calculate the grain size of the synthesized products. And XRD can do qualitative phase analysis because "There is a unique set of interplanar spacing and relative intensities or a distinct XRD pattern to every crystalline mineral (or solid) which serves as its fingerprint . Hence,  XRD is used for determining diffraction intensities and peak positions of a specimen while Bragg’s law is used for  calculating corresponding interplanar spacings.     Therefore,  constituent phases can be identified by computerized searching in an unknown sample. It can be proceeded by matching  thestrongest peak intensity line with recorded interplanar spacing values stored in a database,  namely the joint committee on powder diffraction standard (JCPDS),  while using peak intensity of the strongest line as a function of the weight of phases in a mixture can be used for  phase quantification . Scientific investigations have revealed that in addition to pure mineral composition,  XRD can also provide data about the proportions of minerals in a mixture . XRD produces superior information in  qualitative, quantitative,  and structural analyses of crystalline minerals. It performs well with phases proportions in minerals.  Its full  spectrumfitting method of Rietveld refinement can be used for revealing structural and content information of all phases present in the sample mixture "(please see doi: 10.3390/min12020205).Therefore, we believe that our phase analysis can prove that the synthesized product is hexagonal ZnO, and the SEM image of the product can also provide strong support from the other side.   We hope you ' ll give it more thought and that you ' ll find it suitable for publication.

Reviewer 3 Report

Comments and Suggestions for Authors

The authors have done a good job and followed the majority of reviewers comments and provided answers to relevant questions. I suggest the paper to be published as submitted.

Comments on the Quality of English Language

The English language is now good, as far as I am qualified to assess it.

Author Response

Thank you very much for your support and encouragement to our work. Each of your comments has an important guiding significance for our work, so that we can continuously improve and perfect.

Round 3

Reviewer 2 Report

Comments and Suggestions for Authors

As the authors claim, supercritical hydrothermal synthesis of ZnO nanoparticles appears to be a new technology. However, I did not see any novelty in the ZnO itself synthesized in this study. In other words, it is not clear how the ZnO synthesized by the new technology differs from those synthesized by other technologies. This journal is a high-quality journal. Therefore, I believe that further characterization should be performed to show what advantages their technique has in terms of material properties as well as synthesis methods. In the additional response, the authors only described the capabilities of XRD. Only the structural aspects of the synthesized material have been presented, but it is necessary to show how the physical and chemical properties of this material are different from existing materials. If the authors provide further results on this, I will consider it again.

Round 4

Reviewer 2 Report

Comments and Suggestions for Authors

The authors added FT-IR results and more analysis. This revision may be sufficient for publication.